# Astrovirus Infection in Cattle with Nonsuppurative Meningoencephalitis in South Korea

**DOI:** 10.3390/v13101941

**Published:** 2021-09-28

**Authors:** Sook-Young Lee, Jong-Ho Kim, Yoon-Ji Kim, Young-Sik Kim, Su-Gwon Roh, Kyung-Hyun Lee, Heui-Jin Kim, Jae-Ho Shin, Jae-Ku Oem

**Affiliations:** 1Laboratory of Veterinary Infectious Disease, College of Veterinary Medicine, Jeonbuk National University, Iksan 54596, Jeonbuk, Korea; sylee163@gmail.com (S.-Y.L.); kimyoonji102@naver.com (Y.-J.K.); yoksik@naver.com (Y.-S.K.); 2Animal Disease Diagnostic Division, Animal and Plant Quarantine Agency, 117 Hyeoksin 8-ro, Kimcheon-si 39660, Gyeongsanbuk-do, Korea; whdgh2339@korea.kr (J.-H.K.); sujza26@naver.com (S.-G.R.); mylovehyun@korea.kr (K.-H.L.); jennykim@korea.kr (H.-J.K.); 3Department of Applied Biosciences, Kyungpook National University, 80 Daehak-ro, Daegu 41566, Korea; jhshin@knu.ac.kr

**Keywords:** astrovirus, encephalomeningitis, nonsuppurative, neurological disease, bovine

## Abstract

Neurological diseases in cattle can be caused by several infectious agents. Astroviruses are increasingly recognized as the causative agent of encephalitis in various animals, including humans. In this study, a neuroinvasive astrovirus (BoAstV 20B05) was discovered in the brain tissues of an 81-month-old Korean native cattle with neurological symptoms. Lymphocyte infiltration and multifocal perivascular cuffing were observed in the cerebrum and brain stem, and viral antigens were also detected in the meninges. In particular, the concentration of the astroviral genome was high in the brain tissues. Korean BoAstV 20B05 was classified into the CH13/NeuroS1 clade and was closely related to the Neuro-Uy and KagoshimaSR28-462 strains. Our evolutionary analysis showed that Korean BoAstV 20B05 belongs to the sub-lineage NeuroS1 and evolved independently of BoAstV KagoshimaSR28-462. These results suggest that neuroinvasive astroviruses were first introduced in Korea. However, analysis is limited by the lack of reference astrovirus sequences reported in various countries within Asia, and further analysis should be performed using more strains. In this study, we identified a neuroinvasive astrovirus infection with neurological symptoms for the first time in South Korea and confirmed that BoAstV 20B05 may have been introduced in South Korea a long time ago.

## 1. Introduction

Astroviruses are single-stranded positive-sense RNA viruses that contain a genome of approximately 6.2–7.7 kb in length. Their genome consists of three open reading frames (ORFs) designated as ORF1a, ORF1b, and ORF2 [1]. ORF1ab is located towards the 5′ end of the viral RNA, and it representatively encodes the RNA-dependent RNA polymerase (RdRP) and a viral protease; meanwhile, ORF2 is located toward the 3′ end of the RNA, and it encodes the viral capsid [2]. Astroviruses are classified into two genera: *Mamastrovirus* and *Avastrovirus*, which infect mammalian hosts and avian hosts, respectively [3]. The most common symptoms are diarrhea and gastrointestinal symptoms in mammalian and avian species, as well as in ovine, porcine, feline, mink, and turkey species [2]. Astroviruses have diverse genetic composition that could be due to several ecological and evolutionary processes, such as cross-species transmission and recombination. Moreover, astroviruses have wide host ranges [3,4].

Astroviruses are traditionally known to cause gastrointestinal disease [5]; however, since 2010, several novel strains have been isolated from the brain tissues or cerebral spinal fluid of humans and animals with neurological disease and encephalitis [6]. In 2010, a neuroinvasive astrovirus was first reported in a young boy with X-linked agammaglobulinemia deficiency disease and isolated in mink with shaking mink syndrome using viral metagenomics [7,8]. Subsequently; neuro-astrovirus was identified in cattle with neurologic disease [9]; domestic sheep with encephalitis and ganglionitis [10]; and weaned pigs with encephalomyelitis, weakness, and paralysis [11]. Notably, neuro-astrovirus strains have been identified in bovines in many countries including the United States (BoAstV NeuroS1), Switzerland (BoAstV CH13 and BoAstV CH15), Germany (BoAstV BH89/14), Uruguay (BoAstV Neuro-Uy), Japan (BoAstV KagoshimaSR28-462), and Italy (BoAstV PE3373/2019/Italy) [9,12,13,14,15,16]. Recently, new cases of bovine astrovirus infection with nonsuppurative encephalitis have been reported in Canada [17,18]. To date, detection of astrovirus has been reported in bovine with diarrhea in South Korea [19], but no astrovirus infection cases with neurological signs, type, or symptoms have been reported. This study is the first to identify an astrovirus infection in cattle with neurological symptoms in South Korea.

## 2. Materials and Methods

### 2.1. Samples

An 81-month-old Korean native cattle died 10 days after showing an unknown cause of astasia, paralysis, depression, and ataxia at a cattle farm in Uiseong city, Gyeonsang-buk do, South Korea in March 2020. The cerebrum, cerebellum, and mandibular lymph nodes of this cattle were collected. Samples were tested for the presence of infectious agents of neurological diseases, including bovine diarrhea disease virus, bovine herpesvirus, Aino virus, Ibaraki virus, Akabane virus, Chuzan virus, bovine ephemeral fever virus, and *Listeria monocytogenes* using RT-PCR; however, all samples were negative for these infectious agents.

### 2.2. RNA Extraction, PCR, Sequencing, and RT-qPCR

10% (*w*/*v*) of the cerebrum, cerebellum, and mandibular lymph nodes were ground in 1mL of phosphate-buffered saline (PBS) containing 1% antibiotic-antimycotic solution (Corning, Manassas, VA, USA). Total RNA was extracted from the ground tissue supernatant at a final volume of 200 µL using the QIAamp Viral RNA Mini Kit (Qiagen, Strasse, Hilden, Germany) and finally eluted in 70 µL elution buffer. Random cDNA was synthesized using the SuperScript III First-Strand Synthesis Kit (Invitrogen, Carlsbad, CA, USA), following the manufacturer’s instructions. The RNA-dependent RNA polymerase gene (*RdRp*) of astrovirus was amplified using previously reported primer sets [20], and extension of the genome sequences was performed using specifically designed primer sets (Appendix A). RT-PCR was performed as follows using TaKaRa Ex Taq (Takara, Kusatsu, Shiga, Japan): initial incubation at 94 °C for 10 min; followed by 40 cycles consisting of denaturation at 95 °C for 1 min, annealing at 50–58 °C for 1 min, and extension at 72 °C for 1 min; and a final cycle at 68 °C for 5 min, using Mastercycler (Eppendorf, Hamburg, Germany). PCR products of *RdRp* (440 bp) and extended PCR products were purified using Qiaquick PCR Purification Kit (Qiagen) and sequenced using BigDye^®^ 1.1 terminator cycle sequencing reagents performed in ABI PRISM 3130 Automated Capillary DNA sequencer (Applied Biosystems, Foster City, CA, USA).

In addition, 50 ng of total RNA extracted from each tissue of Korean cattle was used to target ORF2 (4310–4406 nucleotide [nt]) of the astroviral genome using RT-qPCR (Appendix A). The reaction was conducted using TOPreal™ One-step RT-qPCR Kit (Enzynomics, Daejon, South Korea) performed in CFX Real-Time (Bio-Rad, Hercules, CA, USA). RT-qPCR was performed as follows: 1 cycle at 50 °C for 30 min; followed by 45 cycles at 95 °C for 10 min, at 95 °C for 5 s, and at 60 °C for 30 s.

### 2.3. Pathological Examination and In Situ Hybridization (ISH)

The cerebrum, cerebellum, and mandibular lymph nodes were fixed using 10% *v*/*v* neutral buffered formalin and embedded in paraffin wax (FFPE), all sectioned at 4 µm thickness. The FFPE material was stained with hematoxylin and eosin according to standard protocols. Furthermore, fluorescent ISH was performed using a probe targeting ORF2, which is located at 4129–4698 nt (Appendix A). The probe was labeled with digoxigenin during in vitro transcription (PCR DIG Probe Synthesis Kit, Roche, Mannheim, Germany). ISH was performed using an automated system (NexES IHC instrument; Ventana Medical System, Inc., Oro Valley, AZ, USA) and DAB detection system (Ventana Medical Systems Inc., Oro Valley, AZ, USA). The tissue sections were deparaffinized (standard xylene and industrial methylated spirits) and fixed in 4% paraformaldehyde for 10 min. Tissues were then permeabilized by incubation in 1 mg/mL pepsin in 0.1 M HCl for at 37 °C for 20 min at, followed by denaturation at 70 °C for 10 min, before hybridization with DIG-labeled riboprobe at 65 °C for 6h. Subsequently, the sections were washed twice using 0.1 X SSC at 75 °C for 6 min, followed by PBS at room temperature for 5 min. For visualization, the sections were incubated with an anti-digoxigenin-fluorescein antibody (Invitrogen) for 30 min, rinsed twice with PBS for 5 min, and then dried. All tissue sections were covered with slips using 90% (*v*/*v*) glycerol/PBS mounting solution. Then, the DAB detection system was employed, and the sections were incubated at 37 °C. Sections were counterstained with hematoxylin and then counterstained with a bluing reagent.

### 2.4. Genetic Evolutionary Analysis

Phylogenetic analysis was performed based on the obtained partial ORF1 and ORF2 sequences. Available reference sequences were retrieved from GenBank for alignment with Korean bovine astrovirus (accession no. MZ475060), and multiple sequence alignments were generated using the Clustal W algorithm in BioEdit version 7.0.9.0. Phylogenetic trees were generated using Maximum Likelihood in MEGA 7.0 software [21]. In addition, the Bayesian phylogenetic analysis was carried out with the BEAST v1.8.4 package [22] using the obtained coding region of CH13/neuroS1 clade astroviruses to determine the evolutionary rate, the ages/years of the common ancestors, and the most probable route of viral circulation by country. The temporal structure of each dataset was evaluated using TempEst (Edingurgh, UK) [23]. The lognormal relaxed molecular clock with Bayesian Skyline analysis was selected by Bayes factor among the different combinations of molecular clocks and coalescent tree priors used. The Markov chain Monte Carlo length was 100 million generations, ensuring that the posterior probability was used to evaluate clades. The maximum clade credibility tree was obtained using TreeAnnotator software from BEAST and visualized in FigTree v1.4.3.

## 3. Results

### 3.1. Detection of Bovine Astrovirus

The astrovirus *RdRp* gene was detected in all samples. In the cerebellum, forebrain, and midbrain, the genome was detected as the first RT-PCR product (440 bp). In the hindbrain and lymph nodes, *RdRp* was detected after the second RT-PCR (392 bp). Genome extensions using specific primer sets were performed in the forebrain samples, resulting in gene fragments located at 486–1386 nt and 1932–5992 nt. Genome fragments of approximately 400–1100 bp were obtained from the other samples (Table 1).

Detection of the partial ORF2 gene using RT-qPCR was performed in each sample. The results showed that cycle threshold values were approximately 21–22 in the cerebellum, forebrain, and midbrain, and >25 in the hindbrain and lymph nodes. The amounts of quantified viral genomes from the cerebellum, forebrain, midbrain, hindbrain, and lymph nodes were 5.37E6, 1.17E7, 1.07E6, 3.09E6, and 3.09E4, respectively.

### 3.2. Pathological and ISH Anlaysis

All brain samples were negative for the presence of the major viral agents responsible for the neurological manifestations of cattle. Histological examination revealed nonsuppurative meningoencephalitis affecting the cerebrum, brainstem, and cerebellum. Mononuclear cell infiltration, including lymphocytes, plasma cells, and macrophages, was present in the cerebrum (Figure 1a,b). Multifocal perivascular cuffing with many layers of mononuclear cells was also observed in the brainstem (Figure 1c). In the cerebellum, weak infiltration of lymphocytes was detected in the meninges.

### 3.3. Phylogenetic and Evolutionary Analysis

The partial sequences of ORF1ab and ORF2 obtained from the forebrain of 20B05 cattle were phylogenetically analyzed. The *RdRp* (2274–2632 nt; 359 bp) and partial ORF2 (4157–5370 nt; 1214 bp) sequences of Korean bovine astrovirus were classified into the CH13/NeuroS1 group (Figure 2). In addition, the *RdRp* of Korean bovine astrovirus showed the highest sequence similarity with the previously reported Neuro-Uy strain (93.6%) identified from Uruguayan cattle, followed by the Kagoshima SR28-462 strain (93.3%) discovered in Japanese cattle. In addition, the gene was observed to have sequence similarity range of 88.3–91.2% with CH13/Neuro S1 strains. Meanwhile, the ORF2 of Korean bovine astrovirus showed the highest sequence similarity with the Kagoshima SR28-462 strain (94.4.1%), followed by Neuro-Uy strain (93.8%). The ORF2 of Korean bovine astrovirus showed sequence similarity range of 90.1–92.3% with other CH13/Neuro S1 strains. Additionally, the astroviral sequences detected from each brain tissue of the 20B05cattle were identical. However, the ORF1a and ORF2 sequences was observed to have a sequence similarity range of 65.1–72.2% with the neuroinvasive BoAstV CH15 strain, which belongs another phylogenetic group.

Our Maximum clade credibility tree (MCCT) showed that the neuroinvasive astroviruses within the CH13/NeuroS1 clade were classified into two sub-lineages (CH13 and NeuroS1) with a common ancestor (Figure 3). Bovine AstV 20B05 was included in sub-lineage NeuroS1 and diverged from a common ancestor in approximately 1914 (95% highest probability density [HPD] 1843–1975). However, analysis using the partial *RdRp* sequence indicated that the sub-lineage NeuroS1 strains were divided into the North America and Japan group and the South America and Korea group in approximately 1887 (95% HPD, 1748–1996) by high posterior probability value (*p* = 0.89) (Appendix A).

## 4. Discussion

We identified a BoAstV strain, BoAstV 20B05, in the brain tissues of 81-month-old cattle that presented with neurological symptoms. Our histopathological findings mostly observed infiltration of lymphocytes and multifocal perivascular cuffing in the cerebral meninges and brain stem, respectively. In addition, astroviral gene in the meninges were detected using ISH. The Korean BoAstV infection case was similar to other cases of bovine astrovirus-related encephalitis; however, histopathological lesions were limited to the meninges and brainstem. These results could be attributed to the different stages of infection in the central nervous system (CNS) [24]. In practice, bovine nonsuppurative encephalitis is known to manifest variation in the intensity of perivascular cuffs, gliosis, and neuronal necrosis throughout the brain and brainstem [25]. Interestingly, the astroviral genome was detected in brain tissues after the first RT-PCR reaction, and the molecular copy numbers of the viral genome were also high in the brain tissues (copy number > 1 × 10^6^). High concentrations of the neuroinvasive astroviruses in the CNS tissues (brainstem and spinal cord) were detected previously [10,11,12,13,14], and high concentrations of the Korean BoAstV genome were also observed in the cerebellum, forebrain, and midbrain.

Due to its temporary detection in fecal and nasopharyngeal samples, BoAstV CH13-NeuroS1 is assumed to shed through the fecal route [11,26]. As a result, it is considered that the potential reservoirs of astroviruses are the gastrointestinal and respiratory tracts. Although the intestine or other organ tissues were not tested due to the lack of samples in our study, the genetic relationships between gastrointestinal astrovirus and neuroinvasive astrovirus from one individual will provide important clues for determining the infection route of the virus.

Korean BoAstV 20B05 was classified into the CH13/NeuroS1 clade within the *Mamastrovirus* by phylogenetic analysis of partial ORF1ab and ORF2. In particular, BoAstV 20B05 was genetically related to previously reported BoAstV Neuro-Uy and BoAstV KagoshimaSR28-462 identified in Uruguay and Japanese cattle, respectively [14,15]. In ORF1ab, the pairwise distances between BoAstV 20B05 and BoAstV Neuro-Uy and BoAstV Kagoshima SR28-462 were not significantly different, but in ORF2, a higher genetic relationship with the KagoshimaSR28-462 strain was observed. However, no genetic recombination of the BoAstV 20B05 sequence between BoAstV Neuro-Uy and BoAstV Kagoshima SR28-462 was detected. The partial ORF2 sequence of Korean BoAstV 20B05 was observed to be only 1214 bp in length within the conserved region of ORF2. As the genetic recombination and phylogenetic analysis of BoAstV 20B05 did not include the sequence of the hypervariable region of ORF2 [3,27], further analysis is needed, including the sequence of this region.

In cattle, neurotropic astrovirus was first reported in the United States in 2013 [12]. Since then, it has been reported in Europe, Japan, Canada, and Uruguay. This study is the first to report neurotropic astrovirus in Korea. Bovine astroviruses were phylogenetically classified into two clades, CH15 (CH15 and BH89/17) and CH13/NeuroS1 (CH13 strains, NeuroS1, Neuro-Uy, KagoshimaSR28-462, and 20B05) clades, which were reported in Europe and America/Asia, respectively. Our evolutionary analysis showed that the astrovirus belonging to sub-lineages NeuroS1 and CH15 diverged around 1887 from a common ancestor, and since then, the strains within the NeuroS1 clade were spread to America and Asia [15]. In particular, the Japanese strain BoAstV KagoshimaSR28-462 was discovered in 2016 [14], and four years later, the BoAstV 20B05 strain was detected in South Korea. However, 20B05 belonging to sub-lineage NeuroS1 is believed to have evolved independently of BoAstV KagoshimaSR28-462 by divergence from the co-ancestors in approximately 1914. These results suggest that the neuroinvasive astrovirus was first introduced to Korea and then spread to other countries. However, the results are limited by the lack of reference astrovirus sequences reported in various countries within Asia, and further analysis using more strains should be performed. Furthermore, considering that the astrovirus was introduced in 1914 in South Korea, it is thought that a retrospective study of cattle with neurological symptoms is necessary.

## Figures and Tables

**Figure 1 viruses-13-01941-f001:**
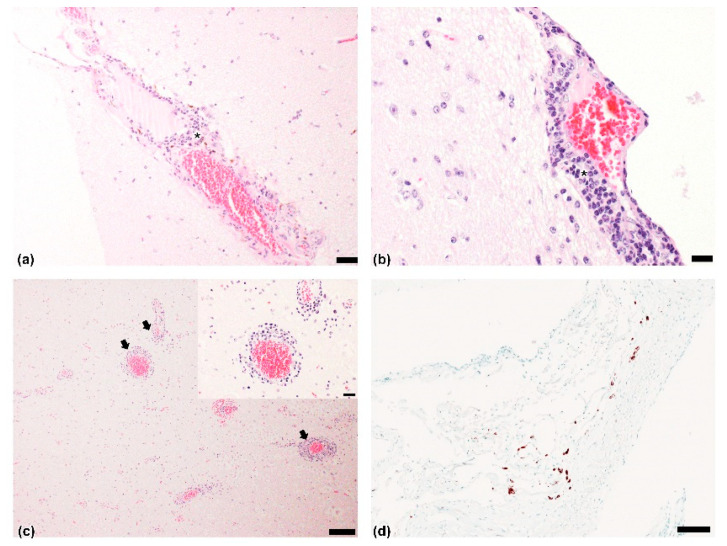
Histopathological findings and in situ hybridization (ISH) results: Cerebrum. Nonsuppurative meningoencephalitis with infiltration of mostly lymphocytes (asterisk) around blood vessels of meninges. Hematoxylin and eosin (H&E). Scale bar = (**a**) 50 µm, (**b**) 20 μm; (**c**) Nonsuppurative encephalitis with multifocal perivascular cuffing (black arrows) with lymphocytes. Brainstem. H&E. Scale bar = 100 µm; insert magnification of perivascular focusing including 4–5 layers of mononuclear cells. H&E. Scale bar = 20 µm; (**d**) Brownish-black antigens observed after labeling of bovine astrovirus in meningeal lymphocytes. Brain. ISH. Scale bar = 100 µm.

**Figure 2 viruses-13-01941-f002:**
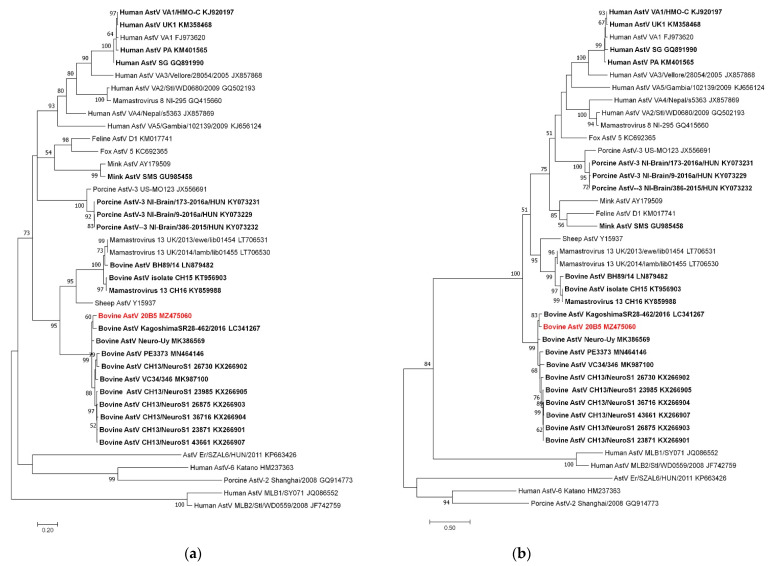
Phylogenetic analysis using ORF1 and ORF2 sequences of bovine astrovirus 20B05: (**a**) Maximum likelihood (ML) tree of ORF1 region (2274–2632 nt; 359 bp); (**b**) ML tree of ORF2 region (4157–5370 nt; 1214 bp). Neurotropic astrovirus strains are indicated in bold letters, and Korean bovine astrovirus is written in red.

**Figure 3 viruses-13-01941-f003:**
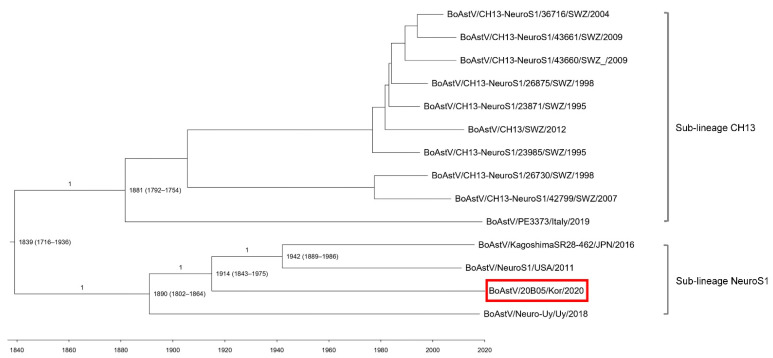
Maximum clade credibility tree generated using the analysis of concatenated open reading frames (partial *RdRp* and ORF2) sequences of BoAstV 20B05 (red box). The posterior probability values are shown in the branches, and the years of origin for each clade with the 95% highest probability interval are represented in each node.

**Table 1 viruses-13-01941-t001:** Detection of astroviral *RdRp* from brain tissues of 20B05 cattle.

Brain Tissue	RT-PCR ^1^	Position (Nucleotide)	Cycle Threshold Value	Copy Number
Cerebellum	+ (+)	2677–3791	22.53	5.37E6
Forebrain	+ (+)	486–1386, 1932–5992	21.34	1.17E7
Midbrain	+ (+)	2708–3797	21.70	9.33E6
Hindbrain	− (+)	3400–3788	25.04	1.07E6
Lymph node	− (+)	3403–3788	30.53	3.09E4

^1^ First RT-PCR (second RT-PCR).

## Data Availability

The data presented in this study are openly available in NCBI database by accession no. MZ475060.

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
