# Peer review of "Astrovirus Infection in Cattle with Nonsuppurative Meningoencephalitis in South Korea"

_viruses, 2021, doi:10.3390/v13101941_

Round 1
Reviewer 1 Report
A very interesting and well written case report on diagnosis of bovine astrovirus infection with nonsuppurative meningoencephalitis for the first time in South Korea.
Please, find my specific comments below.
Introduction:
Page 2 line 51-57: the authors forgot to insert that astrovirus has also been described in Italy: Zaccaria et al. Detection of Astrovirus in a Cow with Neurological Signs by Nanopore Technology, Italy Viruses. 2020 May; 12(5): 530. Published online 2020 May 11. doi:10.3390/v12050530. It would be appropriate to insert it.
Material and Methods:
Page 2 line 66-69: the bovine samples were tested for some neurothropic virus. Were the samples also tested for bacteria and parasites?
Results page 4 line 154 Figure 1
The histological image quality can be improved, in particularly brightness and contrast.
Figure 1 legend:
Delete line 155 the word results. Fig 1a: I think that the image shows a brain vessel and not meningeal vessel, therefore I suggest to change “meningitis” with “meningoencephalitis”
I suggest reversing the order of the words: Cerebrum. Non-suppurative meningoencephalitis…...
References page 7 line 271 change “Benedictis, P.D. with De Benedictis P.
Author Response
Reviewer #1:
Comments to the Author
A very interesting and well written case report on diagnosis of bovine astrovirus infection with nonsuppurative meningoencephalitis for the first time in South Korea.
Please, find my specific comments below.
Introduction:
Page 2 line 51-57: the authors forgot to insert that astrovirus has also been described in Italy: Zaccaria et al. Detection of Astrovirus in a Cow with Neurological Signs by Nanopore Technology, Italy Viruses. 2020 May; 12(5): 530. Published online 2020 May 11. doi:10.3390/v12050530. It would be appropriate to insert it.
RESPONSE: We added the reference in line 57.
Material and Methods:
Page 2 line 66-69: the bovine samples were tested for some neurothropic virus. Were the samples also tested for bacteria and parasites?
RESPONSE: For differential diagnosis of meningoencephalitis, we checked representative bacterial (Listeria monocytogenes) pathogens reported in Korean cow. We aseptically inoculated brain tissue culture swabs onto sheep blood agar and MacConkey agar. Furthermore, additional PCR was carried out to detect L. monocytogenes (hly gene) from the brain tissue. We couldn’t isolated any pathogenic bacteria and PCR result was negative.
Results page 4 line 154 Figure 1
The histological image quality can be improved, in particularly brightness and contrast.
RESPONSE: We changed the brightness and contrast of the figure 1.
Figure 1 legend:
Delete line 155 the word results. Fig 1a: I think that the image shows a brain vessel and not meningeal vessel, therefore I suggest to change “meningitis” with “meningoencephalitis”
I suggest reversing the order of the words: Cerebrum. Non-suppurative meningoencephalitis…...
RESPONSE: We corrected the words in line 161.
References page 7 line 271 change “Benedictis, P.D. with De Benedictis P.
RESPONSE: We corrected the reference.
Reviewer 2 Report
Astroviruses were previously discovered to cause moderate gastrointestinal problem in most animals, but since 2010, they have been increasingly recognized as a neurotropic pathogen in a variety of mammalian species. In the United States, Switzerland, Japan, Uruguay, and Germany, bovine astrovirus-associated encephalitis has been found already. The global prevalence of neurotropic astroviruses in cattle, on the other hand, is still unknown.
Lee and colleagues report the first astrovirus-associated meningoencephalitis in cattle in South Korea in their paper "Astrovirus infection in cattle with nonsuppurative meningoencephalitis in South Korea." They were able to amplify a portion of the astrovirus genomic sequence and phylogenetically analyze its ORF1, ORF2 genes, in addition to proving the existence of viral RNA in tissue samples using ISH. It would be much more beneficial for understanding the pathogenesis of the astroviruses if the author could cultivate the BoAstV 20B05 virus in vitro.
The investigations and analyses are meticulously carried out, and the identification of neurotropic astrovirus in bovines in South Korea adds to the epidemic's knowledge. However, I do have a few concerns that must be addressed.
- In Lines 66–69, there is no information on the type of assays used to detect neurologically related pathogens in the collected samples. It would be more convenient if it were included in the text.
- There is no mention of the amount of tissues sample and PBS used for RNA extraction in lines 71 to 71. However, because these numbers are critical for RNA quantification, please include them in the text.
- In Section 2.2, please specify the RT-PCR reagents used for astrovirus genome RT-PCR.
- The description of “Sections were counterstained with hematoxylin and counterstained with a bluing reagent” in Line 111. is difficult to understand. Have you done two counterstainings or just one with two steps? Please make it clear.
- In Table 1, how much tissue sample is used to determine the copy number of RdRp cDNA or RNA? Also, the table title should indicate that the RdRp is specific to Astroviruses.
- The legend in Figure 1 is poorly described and difficult to understand. Each assay in the experiments should be clearly identified in the legend.
- In line 198, the authors state incorrectly, "In addition, viral antigens in the meninges were detected using ISH." Because ISH detects the virus's RNA rather than its protein; this should be referred to as the antigen gene or simply the viral gene.
Author Response
Reviewer #2:
Astroviruses were previously discovered to cause moderate gastrointestinal problem in most animals, but since 2010, they have been increasingly recognized as a neurotropic pathogen in a variety of mammalian species. In the United States, Switzerland, Japan, Uruguay, and Germany, bovine astrovirus-associated encephalitis has been found already. The global prevalence of neurotropic astroviruses in cattle, on the other hand, is still unknown.
Lee and colleagues report the first astrovirus-associated meningoencephalitis in cattle in South Korea in their paper "Astrovirus infection in cattle with nonsuppurative meningoencephalitis in South Korea." They were able to amplify a portion of the astrovirus genomic sequence and phylogenetically analyze its ORF1, ORF2 genes, in addition to proving the existence of viral RNA in tissue samples using ISH. It would be much more beneficial for understanding the pathogenesis of the astroviruses if the author could cultivate the BoAstV 20B05 virus in vitro.
The investigations and analyses are meticulously carried out, and the identification of neurotropic astrovirus in bovines in South Korea adds to the epidemic's knowledge. However, I do have a few concerns that must be addressed.
- In Lines 66–69, there is no information on the type of assays used to detect neurologically related pathogens in the collected samples. It would be more convenient if it were included in the text.
RESPONSE: We added the method for detection of infectious agents in line 71.
- There is no mention of the amount of tissues sample and PBS used for RNA extraction in lines 71 to 71. However, because these numbers are critical for RNA quantification, please include them in the text.
RESPONSE: We added the amount of tissue samples and PBS in line 73-74.
- In Section 2.2, please specify the RT-PCR reagents used for astrovirus genome RT-PCR.
RESPONSE: We added information on RT-PCR reagent in line 82.
- The description of “Sections were counterstained with hematoxylin and counterstained with a bluing reagent” in Line 111. is difficult to understand. Have you done two counterstainings or just one with two steps? Please make it clear.
RESPONSE: We have done two counterstainings. We corrected in line 114-115.
- In Table 1, how much tissue sample is used to determine the copy number of RdRp cDNA or RNA? Also, the table title should indicate that the RdRp is specific to Astroviruses.
RESPONSE: We have been described the RNA concentration used for RT-qPCR in line 89. Also, as your opinion, we corrected the title of table 1.
- The legend in Figure 1 is poorly described and difficult to understand. Each assay in the experiments should be clearly identified in the legend.
RESPONSE: We corrected in Figure 1.
- In line 198, the authors state incorrectly, "In addition, viral antigens in the meninges were detected using ISH." Because ISH detects the virus's RNA rather than its protein; this should be referred to as the antigen gene or simply the viral gene.
RESPONSE: We corrected in line 204.
Round 2
Reviewer 1 Report
Comments to the Author
A very interesting and well written case report on diagnosis of bovine astrovirus infection with nonsuppurative meningoencephalitis for the first time in South Korea.
Please, find my specific comments below.
Introduction:
Page 2 line 51-57: the authors forgot to insert that astrovirus has also been described in Italy: Zaccaria et al. Detection of Astrovirus in a Cow with Neurological Signs by Nanopore Technology, Italy Viruses. 2020 May; 12(5): 530. Published online 2020 May 11. doi:10.3390/v12050530. It would be appropriate to insert it.
RESPONSE: We added the reference in line 57.
QUESTION: As didn’t add the suggestion in philogenetyc analysis, at least discussed it in the light of their genetic relatedness e.g. percent of nucleotides and aminoacidic identify of astrovirus detected in Italy.
Material and Methods:
Page 2 line 66-69: the bovine samples were tested for some neurothropic virus. Were the samples also tested for bacteria and parasites?
RESPONSE: For differential diagnosis of meningoencephalitis, we checked representative bacterial (Listeria monocytogenes) pathogens reported in Korean cow. We aseptically inoculated brain tissue culture swabs onto sheep blood agar and MacConkey agar. Furthermore, additional PCR was carried out to detect L. monocytogenes (hly gene) from the brain tissue. We couldn’t isolated any pathogenic bacteria and PCR result was negative.
QUESTION: Neospora caninum and Toxoplasma gondii were also tested?
Results page 4 line 154 Figure 1
The histological image quality can be improved, in particularly brightness and contrast.
RESPONSE: We changed the brightness and contrast of the figure 1.
QUESTION: The histological image quality wasn’t improved as suggested, the image in the revised text are the same. Please improve it as possible.
Figure 1 legend:
Delete line 155 the word results. Fig 1a: I think that the image shows a brain vessel and not meningeal vessel, therefore I suggest to change “meningitis” with “meningoencephalitis”
I suggest reversing the order of the words: Cerebrum. Non-suppurative meningoencephalitis…...
RESPONSE: We corrected the words in line 161.
QUESTION: the terms “meningitis” as “meningeal” are uncorrected, I suggest” meningoencefalitis”, since meningeal and brain vessels are both involved.
References page 7 line 271 change “Benedictis, P.D. with De Benedictis P.
RESPONSE: We corrected the reference.
Author Response
Dear Dr. Eric O. Freed
Thank you for inviting us to submit a revised draft of our manuscript entitled, "Astrovirus infection in cattle with nonsuppurative meningoencephalitis in South Korea" to Viruses. We also appreciate the time and effort you and each of the reviewers have dedicated to providing insightful feedback on ways to strengthen our paper. Thus, it is with great pleasure that we resubmit our article for further consideration. We have incorporated changes that reflect the detailed suggestions you have graciously provided. We also hope that our edits and the responses we provide below satisfactorily address all the issues and concerns you and the reviewers have noted.
To facilitate your review of our revisions, the following is a point-by-point response to the questions and comments.
Reviewer #1:
Comments to the Author
A very interesting and well written case report on diagnosis of bovine astrovirus infection with nonsuppurative meningoencephalitis for the first time in South Korea.
Please, find my specific comments below.
Introduction:
Page 2 line 51-57: the authors forgot to insert that astrovirus has also been described in Italy: Zaccaria et al. Detection of Astrovirus in a Cow with Neurological Signs by Nanopore Technology, Italy Viruses. 2020 May; 12(5): 530. Published online 2020 May 11. doi:10.3390/v12050530. It would be appropriate to insert it.
RESPONSE: We added the reference in line 57.
QUESTION: As didn’t add the suggestion in philogenetyc analysis, at least discussed it in the light of their genetic relatedness e.g. percent of nucleotides and aminoacidic identify of astrovirus detected in Italy.
RESPONSE: We added the results in Phylogenetic and evolutionary analysis section and supplementary figure.
Material and Methods:
Page 2 line 66-69: the bovine samples were tested for some neurothropic virus. Were the samples also tested for bacteria and parasites?
RESPONSE: For differential diagnosis of meningoencephalitis, we checked representative bacterial (Listeria monocytogenes) pathogens reported in Korean cow. We aseptically inoculated brain tissue culture swabs onto sheep blood agar and MacConkey agar. Furthermore, additional PCR was carried out to detect L. monocytogenes (hly gene) from the brain tissue. We couldn’t isolated any pathogenic bacteria and PCR result was negative.
QUESTION: Neospora caninum and Toxoplasma gondii were also tested?
RESPONSE: Tests on Neospora caninum and Toxoplasma gondii were not performed because they were inconsistent with pathological findings.
Results page 4 line 154 Figure 1
The histological image quality can be improved, in particularly brightness and contrast.
RESPONSE: We changed the brightness and contrast of the figure 1.
QUESTION: The histological image quality wasn’t improved as suggested, the image in the revised text are the same. Please improve it as possible.
RESPONSE: We changed the brightness and contrast of the figure 1.
Figure 1 legend:
Delete line 155 the word results. Fig 1a: I think that the image shows a brain vessel and not meningeal vessel, therefore I suggest to change “meningitis” with “meningoencephalitis”
I suggest reversing the order of the words: Cerebrum. Non-suppurative meningoencephalitis…...
RESPONSE: We corrected the words in line 161.
QUESTION: the terms “meningitis” as “meningeal” are uncorrected, I suggest” meningoencefalitis”, since meningeal and brain vessels are both involved.
RESPONSE: We corrected in line 155.